bioinformatics/microbiology/genomics

metagenomics, strain-tracking, mother–infant, gut microbiome

**Authors for correspondence:**
Hyunmin Koo
e-mail: khmkhm87@uab.edu
Casey D. Morrow
e-mail: caseym@uab.edu

# An individualized mosaic of maternal microbial strains is transmitted to the infant gut microbial community

Hyunmin Koo[1], Braden C. McFarland[2],
Joseph A. Hakim[3], David K. Crossman[1],
Michael R. Crowley[1], J. Martin Rodriguez[4],
Etty N. Benveniste[2] and Casey D. Morrow[2]

[1]Department of Genetics, [2]Department of Cell, Developmental and Integrative Biology, [3]School of Medicine, and [4]Division of Infectious Diseases, Department of Medicine, University of Alabama at Birmingham, Birmingham, AL 35294, USA

HK, 0000-0001-6630-2165; BCM, 0000-0002-7517-9688; DKC, 0000-0002-0981-169X; CDM, 0000-0001-8812-2341

To understand the origins of the infant gut microbial community, we have used a published metagenomic dataset of the faecal microbiome of mothers and their related infants at early (4, 7 and 21 days) and late times (6–15 months) following birth. Using strain-tracking analysis, individual-specific patterns of microbial strain sharing were found between mothers and infants following vaginal birth. Overall, three mother–infant pairs showed only related strains, while 12 infants of mother–infant pairs contained a mosaic of maternal-related and unrelated microbes. Analysis of a second dataset from nine women taken at different times of pregnancy revealed individual-specific faecal microbial strain variation that occurred in seven women. To model transmission in the absence of environmental microbes, we analysed the microbial strain transmission to F1 progenies of human faecal transplanted gnotobiotic mice bred with gnotobiotic males. Strain-tracking analysis of five different dams and their F1 progeny revealed both related and unrelated microbial strains in the mother's faeces. The results of our analysis demonstrate that multiple strains of maternal microbes, some that are not abundant in the maternal faecal community, can be transmitted during birth to establish a diverse infant gut microbial community.

## 1. Introduction

Early studies on the natural history of the infant gut microbiome described extensive short-term changes in the composition during

the first 3 years after birth [1–3]. During this time in the development of the gut microbiome, important microbial interactions are established that are essential for the function of the microbiome community in the digestion of food, metabolism and colonization resistance [4,5]. In recent years, the use of next-generation sequencing methods has evolved to allow a more in-depth examination of the microbial community structure. Using these techniques, genomic variants (i.e. strains) of microbes have now been identified from metagenomic sequencing of complex faecal communities [6–8]. We have previously developed a strain-tracking method called window-based single nucleotide variant (SNV) similarity (WSS) to assess the strain relatedness of multiple microbes in two separate samples [9]. Using the WSS method, a pairwise genome-wide SNV similarity comparison can be performed for a given microbes' relatedness between two samples. Each WSS species' cut-off value was established from our previous study in which the Human Microbiome Project (HMP) dataset was used to distinguish a related strain pair (both strains were taken from the same individual at separate times) from a non-related strain pair (both strains were taken from different individuals) [6,9]. We have applied the WSS analysis to track related strains in individuals following antibiotic(s) treatment, faecal transplant for recurrent *Clostridioides difficile* or gastric bypass surgery that disrupts the physiological environment of the small intestine [9–11].

Studies have used metagenomic sequencing to demonstrate the transfer of maternally related strains to infants following vaginal birth [12–16]. However, these studies have also detected unrelated strains in these infants leading to the speculation that these strains were acquired from the environment [14,17]. In the current study, we have used previously published metagenomic datasets of high-density longitudinal faecal samples from pregnant women, mothers and their infants with our WSS strain-tracking method along with a second strain-tracking method, StrainPhlAn [18,19], to analyse individual-specific faecal microbial strain variation during pregnancy and transmission to infants. Additionally, using gnotobiotic mice transplanted with human faecal samples and bred with gnotobiotic males within a germ-free incubator, we have analysed the transmission of strains from the dams to their F1 progenies in a germ-free environment. Our analysis provides new insights into the origin of microbial strains in the complex infant microbial community.

# 2. Material and methods

## 2.1. Public datasets

We used two publicly available datasets: (i) Shao *et al.* [20] and (ii) Goltsman *et al.* [21] to conduct strain-tracking analysis. For Shao *et al.*, we selected faecal samples from 21 mothers and their infants that mostly included early (4, 7 and 21 days) and late (6–15 months; infancy) time points. As described in Shao *et al.*, faecal samples from the mother were collected in the maternity unit pre- or post-delivery or were collected during delivery by midwives. A subset of 15 mothers and their infants was selected from Shao *et al.*, based on vaginal delivery with the normally observed *Bacteroides* profile (electronic supplementary material, table S1). Additionally, a subset of six mothers and their infants was selected based on Caesarean section delivery with either a low level or normal level of observed *Bacteroides* profile (electronic supplementary material, table S1). For Goltsman *et al.*, we used longitudinal faecal samples collected from nine pregnant women. Within these nine, six women (Term 1–6) gave birth after 37 weeks and three women (Pre 1, 2, and 4) gave birth prematurely. Due to collection days during pregnancy and the number of sequence reads for each collection day varying, we merged each collection day per individual into 1–100, 101–150, 151–200 and 201–300 days to perform strain-tracking analysis (electronic supplementary material, table S1).

## 2.2. Faecal transplantation of gnotobiotic mice

Human faecal samples from five normal healthy volunteers were collected, processed and archived as previously described [22]. These samples were previously used in the UAB faecal transplant programme for patients with recurrent *C. difficile* infection.

Female gnotobiotic mice (10BitFoxP3.GFP.B6, [23]) were individually colonized with the archived five different human faecal samples (*herein* called dam), as previously described [22]. Briefly, each faecal sample was first defrosted on ice and a total of 200 μl of faecal homogenate were delivered via oral gavage. Mice were individually housed in separate isolators in the UAB Gnotobiotic Facility, and sterile (autoclaved) mouse chow (Teklad #2019S) was supplied ad libitum. Previously, we determined that two weeks post-transplant was sufficient to allow re-constitution of an intact microbiome [22].

For the current study, faecal samples from the dams were taken at two weeks post-faecal transplant, and a gnotobiotic male was then added to each female to create an isolated breeder cage. Faecal samples were taken from the F1s ($n = 4$ per each dam) three weeks after birth at the time of weaning (six to seven weeks from the dam sample taken). F1 progenies were females for the donor 3 group and other F1 progenies for the remaining four donor groups (donor 1, donor 2, donor 4 and donor 5) were males. At no time were the dams or F1 progeny removed from the gnotobiotic facility to minimize environmental contamination.

## 2.3. Metagenomic sequencing of murine faecal samples

Faecal samples from individual mice were processed for DNA as previously described [22]. The DNA from a total of 30 faecal samples (5 human faeces, 5 dam faeces and 20 F1 progeny faeces) was prepared, processed and sequenced using the Illumina NextSeq platform, with 150 base paired-end reads with an average of 17 million reads per sample (electronic supplementary material, table S1).

## 2.4. Total sequence reads and processing

A total of 1 354 414 735 metagenomics sequencing reads were downloaded from the two public datasets; 879 152 368 reads from Shao *et al.* and 475 262 367 from Goltsman *et al.* (electronic supplementary material, table S1). Additionally, a total of 539 889 528 sequence reads from the post-faecal microbiota transplant (FMT) gnotobiotic mouse dataset were used for the analysis (electronic supplementary material, table S1). Sequence reads were then filtered to remove adapters, low-quality reads (sliding window of 50 bases having a QScore < 20) and short sequences (sequence length < 50 bases) using Trimmomatic (v. 0.36). Host reads were also filtered by mapping all sequence reads to either hg19 human reference genome or mm10 mouse reference genome using bowtie2 (v. 2.3.4.3), with default parameters [24]. After quality-based trimming and filtering processes, a total of 1 844 185 791 sequences from the three datasets were used for the downstream analyses (electronic supplementary material, table S1).

## 2.5. Strain-tracking analyses

From the mother–infant dataset [20], each mother's sample was only compared with her infant's samples collected at different time points (4, 7, 21 days) and 6–15 months (*herein* called infancy). For the pregnancy microbiome dataset [21], early combined gestational days (1–100 days) were separately compared with any available mid-to-late gestational days (101–150 days; 1–100 versus 151–200 days; 1–100 days versus 201–300 days). Lastly, for the post-FMT gnotobiotic mouse dataset (this study), each human faecal sample was compared with the corresponding dam and F1 progenies; each dam was compared with their F1 progeny.

For the WSS analysis, high-quality processed reads were aligned to the 93 microbial reference sequences [9], which were common and dominant in stool samples collected from healthy European and North American donors [6,9] using the Burrows–Wheeler aligner tool BWA-MEM (v. 0.7.13) with the '-M' option [25]. Mapped reads were then filtered to exclude reads mapped on multiple locations or a low percentage match (less than 90%) using 'mgSNP_sam-filter.py' implemented in the WSS. The filtered reads were sorted and marked for duplicates using Picard Toolkit (v. 1.129, http://broadinstitute.github.io/picard/), and then used for indel realignment using Genome Analysis Toolkit (GATK; v. 3.7) [26]. Multi-sample SNVs for each given reference sequence were measured among all samples for each set using GATK. The resultant multi-sample variant call format (VCF) files were then used to extract SNV information for every possible pair of samples for each microbial species using 'run_mgSNP_cov.sh' code. The resultant VCF file was used for pairwise comparison of the genomic windows (set at 1000 base pairs) determined for each strain using 'mgSNP_compare.sh' code [9]. Any sample having low sequence coverage (less than 30%) and low sequence depth (less than 3.5) against their given reference sequences were excluded from the pairwise comparisons. Also, a low coverage window with more than 50% of the bases having a read depth less than 5 were ignored when comparing the SNV similarity between sample pairs. All codes implemented in the WSS were deposited and are available at https://github.com/hkoo87/mgSNP_2 [9].

After the filtering processes, species that were able to provide the WSS score were selected from each dataset (electronic supplementary material, table S2). To distinguish a related strain pair for each sample pair (i.e. related strain pair between (i) mother–infant, (ii) pregnant woman at different gestational days, (iii) human faecal sample and dam, or (iv) dam and her F1 progenies), a WSS score for each species was compared against each species' cut-off value that was established based on the HMP dataset in our

previous study (related strain pair: WSS score > cut-off; unrelated strain pair: WSS score < cut-off) [9,11]. Species that did not have a cut-off value were excluded (electronic supplementary material, table S2). The analysis for all datasets was summarized and visualized using Microsoft Excel (Microsoft, Seattle, WA, USA).

Strain-tracking analysis for *Bacteroides vulgatus* was additionally performed using StrainPhlAn using default parameters and with the options '-relaxed_parameter3, -marker_in_clade 0.1' [8]. For StrainPhlAn, the high-quality processed reads were mapped against the set of species-specific marker gene database established in MetaPhlAn [18,19]. The sample-specific markers were then reconstructed by using the variant calling approach, and then the reconstructed markers were used to build a phylogenetic tree of the strains [8]. The resultant phylogenetic tree for *B. vulgatus* was visualized using the neighbour-joining method in Jalview using default parameters [27].

## 2.6. Taxonomic profile and $\beta$-diversity

The quality assessed sequence files from the post-FMT gnotobiotic mouse dataset were processed using MetaPhlAn2 (v. 2.7.62) [18], which uses a library of clade-specific markers, to obtain a taxonomic profile for each sample. The resultant taxa abundance table which included 'estimated number of reads from the clade' values was then standardized by Hellinger transformation, and dissimilarities between all pairs of samples were calculated using the Bray–Curtis dissimilarity matrix. Non-metric multidimensional scaling (NMDS) plot was generated to visualize variation between each sample using the 'metaMDS' function in the vegan R package [28]. For better visualization, samples were grouped by ellipses with a confidence interval of 95% using 'ordiellipse' function in the vegan R package [28,29]. A cluster dendrogram was also generated based on the Bray–Curtis dissimilarity matrix using 'hclust (method = average)' function in the vegan R package [28,30]. Permutational multivariate analysis of variance (PERMANOVA) with the function ADONIS in the vegan R package [28] was conducted to compare the microbial community structure on different human faeces groups.

# 3. Results

## 3.1. Strain-tracking between mother and infants

Shao *et al.* [20] described the vertical transmission of faecal microbes from mother to infant. From that study, faecal samples were taken from the mothers and their infants (one sample from each mother and multiple samples from each infant following birth). After the coverage-based filtering process and exclusion of any species that did not have a cut-off value, a total of 17 species that included members of genera *Alistipes*, *Akkermansia*, *Bacteroides*, *Barnesiella*, *Bifidobacterium*, *Collinsella*, *Faecalibacterium*, *Parabacteroides* and *Prevotella* were found in the mothers–infants following vaginal birth (electronic supplementary material, table S2).

A WSS analysis of each species in this study compared against the cut-off value showed the relatedness of the sample pairs at various time points. All pairwise comparisons conducted between each mother's sample and her infant's samples following vaginal birth are shown in the electronic supplementary material, table S3. Overall, *B. vulgatus* was found to be the most abundant species across all mother–infant sample pairs following vaginal birth (electronic supplementary material, table S3). From a comparison of each mother to infancy samples, 7 of the 11 pairs (B02358, B01948, B01194, C01917, C01994, B01364 and C01840) had a related *B. vulgatus* strain (green box in figure 1*a*). By contrast, comparison of 4 of the 11 mother–infancy pairs (A01805, A01763, C01204 and B01719) did not have a related *B. vulgatus* strain (red box in figure 1*a*). We noted similar patterns of related and unrelated mother–infancy pairs for other *Bacteroides* spp. and other microbes. We did also find a unique example in *Bifidobacterium adolescentis* where all five observed mother–infancy pairs were unrelated (red box in figure 1*a*).

To further explore the presence of the unrelated strains in mother–infant pairs, we next compared the WSS scores for each mother and her infant's early time samples (4, 7 and 21 days). Overall, within nine pairs, three of the mother–infant pairs (C01994, C01840 and C01917) had WSS scores above the cut-off from all possible pairwise comparisons, indicating shared strain of *B. vulgatus* observed between mother and her infant at all time points (4, 7, 21 days and infancy). However, comparisons of five of the nine mother–infant pairs (C01204, B01948, B01364, B01719 and C02061) had unrelated *B. vulgatus* strains at certain time points with or without having shared strain at other time points (blue box in

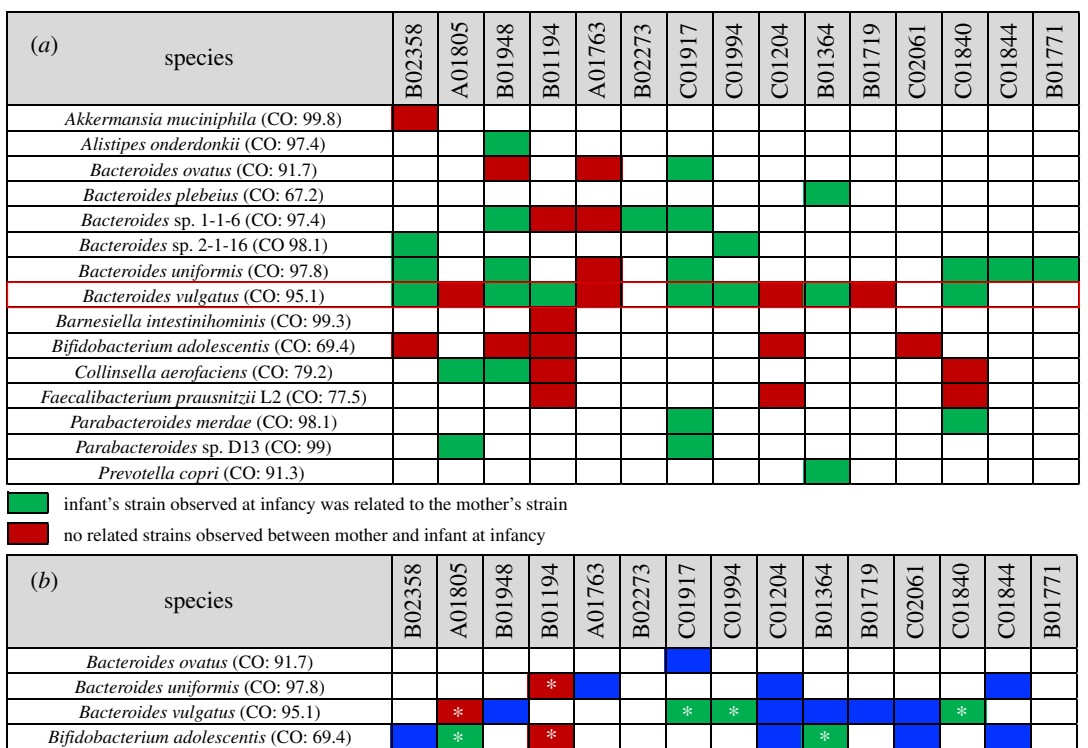

**Figure 1.** Summarized WSS scores between mothers and their infants with vaginal birth. A total of 17 species that had a cut-off value were selected from Shao et al. [20] to compare the WSS scores between (a) each mother's sample and her infant's infancy sample (6–15 months after birth), and (b) each mother's sample and every possible pair of her infant's samples (4, 7, 21 days after birth). The summarized WSS scores of the selected 15 species per each mother and her infant's set were grouped into different colour boxes (colours described in the figure labels). Each column in the table shows an individual ID and matches the number shown in the electronic supplementary material, table S1. WSS scores for all identified species are provided in the electronic supplementary material, table S3. Additional strain profiling analysis was performed for *B. vulgatus* (red outlined box in (a)); result from this analysis shown in electronic supplementary material, figure S1.

figure 1b; electronic supplementary material, table S3). We did also find one mother–infant pair (A01805) that had no shared *B. vulgatus* strains at all time points (red box with an asterisk in figure 1b). We found that the majority of the mother–infant pairs were unrelated at some time points (electronic supplementary material, table S3). We also noted a similar result for *B. adolescentis* and *Collinsella aerofaciens* in which certain mother–infant pairs had related and unrelated strains (figure 1b). In total, for all observed microbial strains, 3 mother–infant pairs (B02273, C01994 and B01771) showed only related strains for all available time points, while 12 infants of mother–infant pairs contained a mosaic of maternal-related and unrelated microbes.

To substantiate the WSS analysis, we have additionally used StrainPhlAn to assess strain relatedness for the mother–infant pairs [8]. Since *B. vulgatus* was found in 12 of the 15 pairs, we selected this species to conduct StrainPhlAn (electronic supplementary material, figure S1). In this analysis, we found five cases (B01948, A01763, B01840, B02358 and B01194) where the WSS and StrainPhlAn agreed, particularly when each mother's sample was compared with her infant's infancy sample (figure 1a; electronic supplementary material, figure S1). StrainPhlAn analysis on the remaining cases showed a partial agreement or disagreement with the WSS (electronic supplementary material, table S3).

Finally, Shao et al. [20] also reported the vertical transmission of faecal microbes from mother to infant following Caesarean section birth. All pairwise WSS comparisons conducted between each mother's sample and her infant's samples following Caesarean section birth are shown in electronic

supplementary material, table S4. From this analysis, we found that *Bacteroides* sp. 2-1-16 and *B. vulgatus* were shared between mother–infant pairs (B00075 and BBS0070, respectively) for a short time after birth (4–7 days), but not found at later times (infancy). By contrast, *Bacteroides cellulosilyticus*, *Barnesiella intestinihominis*, *B.* sp. 2-1-16, and *Parabacteroides* sp. D13 were shared between one mother–infant pair (A02110) with samples available only at infancy, while *B. ovatus* was shared between one mother–infant pair (B01772) only at infancy. These results support that different mother to infant transmission patterns with respect to related and unrelated strains can occur in Caesarean section-derived babies. One caveat is that relatively smaller sample size for the Caesarean section birth compared to the vaginal birth can affect the transmission frequency between mother and infant.

## 3.2. Strain-tracking during pregnancy

One source of new microbial strains in infants could be the transfer of 'transient' maternal microbes that are present in the mother during pregnancy. To address the possibility that strain variation could occur during pregnancy, we have made use of a recent metagenomic dataset of longitudinal faecal samples collected from nine pregnant women before giving birth [21]. After the coverage-based filtering process, a total of 30 species were found in the pregnancy microbiome dataset (electronic supplementary material, table S5). Within 30 species, 6 species were excluded for further analysis due to the absence of the cut-off value (electronic supplementary material, table S5). Analysis of the 24 species in this dataset revealed a composition that was comparable to that seen for the mother–infant pairs. In particular, all nine women contained at least one *Bacteroides* sp. with seven of the nine containing *B. vulgatus* (figure 2). In support that strain variation can occur during pregnancy, we found that two of these seven women had WSS scores in which unrelated *B. vulgatus* was detected during pregnancy. To further validate this result, we used StrainPhlAn [8] to compare the strains of *B. vulgatus* for each woman during pregnancy (figure 3). From this analysis, we found five cases (Pre 1, Pre 2, Term 1, Term 3 and Term 6) where the WSS and StrainPhlAn agreed. We also found that unrelated *B. vulgatus* strains detected by the WSS were also identified as unrelated by StrainPhlAn (figure 3).

## 3.3. Strain-tracking post-FMT gnotobiotic mouse

Although the presence of microbial strain variation in the pregnant women suggests a possible source for the new strains transmitted between mother and infants, we cannot be certain that some of the strain variations are due to environmental rather than maternal transmission. To address this, we have made use of a surrogate in which gnotobiotic female mice were individually transplanted with five different human faecal samples to create five unique humanized microbiome mice. We first compared the microbial community composition at the species level of each dam and corresponding F1 progeny samples using the NMDS plot (figure 4a). This analysis revealed a significant ($R^2 = 0.893$, p-value = $1 \times 10^{-04}$, PERMANOVA) clustering of the dam and corresponding F1 progeny for the five different faecal transplants consistent with the derivation of the dam from five different human faecal samples. Moreover, each of the five different dams and F1 progeny combinations were clustered together based on the cluster dendrogram (figure 4b).

A WSS strain-tracking analysis was then used to discern the relationship between the donor and the corresponding dam and F1 progenies (figure 5). After the filtering process, a total of 20 species were found from the post-FMT gnotobiotic mouse dataset (electronic supplementary material, table S2). Within 20 species, 3 species were excluded for further analysis due to the absence of the cut-off value (electronic supplementary material, table S2). All pairwise comparisons conducted between each donor versus corresponding dams and their F1 progenies were shown in electronic supplementary material, table S6, and comparisons between each female dam versus their F1 progenies are displayed in electronic supplementary material, table S7. The taxa abundance table including the entire microbial community composition across all samples is shown in electronic supplementary material, table S8.

There were four different patterns that were observed between the donors and the corresponding dams and their F1 progenies (figure 5). First, when each donor and dam sample was compared to the corresponding F1 progeny samples, all observed WSS scores were above the cut-off, suggesting strains were transmitted from donor to F1 progenies (green boxes with an asterisk in figure 5). Second, WSS scores were above the cut-off for the comparison between each dam and the F1 progenies only, implying the dam's strains were transmitted to the F1 progenies (green boxes in figure 5). Third, WSS

| species | Pre 1 | Pre 2 | Pre 4 | Term 1 | Term 2 | Term 3 | Term 4 | Term 5 | Term 6 |
|---|---|---|---|---|---|---|---|---|---|
| *Akkermansia muciniphila* (CO: 99.8) | | G* | | G* | | | | | |
| *Alistipes onderdonkii* (CO: 97.4) | G* | | | | | G* | | | |
| *Alistipes putredinis* (CO: 93.9) | G* | G* | | G* | | G* | | | |
| *Alistipes shahii* (CO: 92) | | | | G* | | G* | | | |
| *Bacteroides cellulosilyticus* (CO: 98.2) | R | | | R | | G* | G* | | |
| *Bacteroides massiliensis* (CO: 97.7) | | | | | G* | G* | | | |
| *Bacteroides ovatus* (CO: 91.7) | G | | | R | R | G* | | G* | |
| *Bacteroides* sp. 1-1-6 (CO: 97.4) | G | | | | | | | | |
| *Bacteroides* sp. 2-1-16 (CO 98.1) | | | G* | | | | | | |
| *Bacteroides stercoris* (CO: 97.1) | | | | G* | | | | | G |
| *Bacteroides uniformis* (CO: 97.8) | G* | | | G* | G | R | | G* | R |
| *Bacteroides vulgatus* (CO: 95.1) | G* | G* | | R | G* | G* | G* | | R |
| *Barnesiella intestinihominis* (CO: 99.3) | G* | | | | | | | | |
| *Clostridium* sp. L2-50 (CO: 72.6) | | | | G* | | | | | |
| *Dialister invisus* (CO: 92.7) | | G | | | | | | | |
| *Eubacterium eligens* (CO: 85.8) | G* | G | | G* | | | | | |
| *Eubacterium rectale* (CO: 94.9) | G* | R | | | | | | | |
| *Eubacterium siraeum* (CO: 99.6) | | | | R | | | | | |
| *Faecalibacterium prausnitzii* A2 (CO: 90) | G | | | G* | | | | | |
| *Faecalibacterium prausnitzii* L2 (CO: 77.5) | | G* | | | | G* | | | |
| *Faecalibacterium prausnitzii* SL3 (CO: 81.5) | R* | R* | | | | G* | G* | | |
| *Parabacteroides merdae* (CO: 98.1) | G | | | G* | | | | | |
| *Parabacteroides* sp. D13 (CO: 99) | G | | | G* | | | | | |
| *Prevotella copri* (CO: 91.3) | | G* | | | | | | | |

Legend:
- ▮ (green) related strain observed at a certain day of pregnancy without the appearance of an unrelated strain
- ▮* (green with asterisk) no strain changes during the entire days of pregnancy
- ▮ (red) no related strain observed at a certain day(s) of pregnancy
- ▮* (red with asterisk) no related strain observed during the entire days of pregnancy

**Figure 2.** Summarized WSS scores for pregnancy microbiome dataset. A total of 24 species that had cut-off values were selected from the pregnancy microbiome dataset [21] to compare the WSS scores between early gestational days (1–100 days) and every possible pair of later gestational days (101–150, 151–200 and 201–300 days) per individual. The summarized WSS scores of the 24 species per each individual were grouped into different colour boxes (colours described in the figure labels). Each column in the table represents an individual ID and matches the number shown in electronic supplementary material, table S1. WSS scores for all observed species are provided in electronic supplementary material, table S4. *Bacteroides vulgatus* was selected to conduct an additional strain profiling analysis (red outlined box) and the resultant data shown in figure 3.

scores that were above the cut-off for the comparison between the donor and corresponding F1 progenies only, suggesting the strain found in F1 progenies was transmitted from the donor even though that strain was not detected in the dams (blue boxes in figure 5). Fourth, WSS scores that were all below the cut-off for all pairwise comparison between each donor, dam and F1 progenies, representing a presence of unrelated strains in F1 progenies post-FMT (red boxes in figure 5). We detected maternal transmission of microbes to the F1s in all of the five different faecal transplants (figure 5). Similar to what we observed in the mother–infant dataset of Shao *et al.*, we found the transmission of the genera *Alistipes*, *Akkermansia*, *Bacteroides*, *Barnesiella*, *Parabacteroides* from the dam to F1s. All of the five transplants had *Bacteroides* spp. and *Parabacteroides* spp. with three of the five transplants having F1 progeny with *B. vulgatus*. In particular, we noted that for the donor 2 group, both the HuM2 F1-1 and HuM2 F1-2 were related to the dam, while HuM2 F1-3 and HuM2 F1-4 were both unrelated to the dam or the original donor 2 faecal sample (figure 5*b*). We also noted examples of *B. uniformis* (figure 5*a*) and *B. ovatus* (figure 5*b,c*) where the strains were detected that were not related to the corresponding dam or donor faecal samples. We also noted several instances where the strains in the F1's were only found in the original faecal sample used for transplant (blue boxes in figure 5). Collectively, our results support that a mosaic of maternal strains,

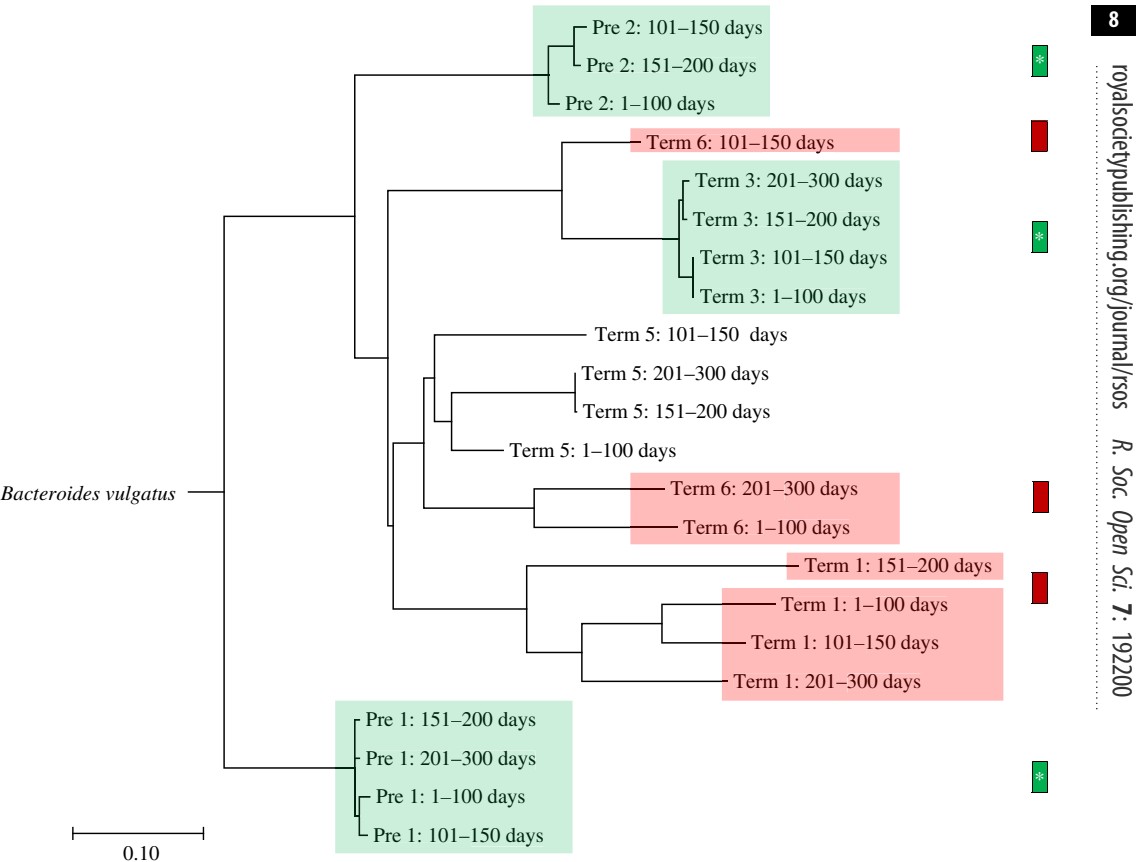

**Figure 3.** StrainPhlAn on the pregnancy microbiome dataset. Strain profiling was additionally performed for *B. vulgatus* using StrainPhlAn across all samples, and 14 samples were able to represent the relatedness of *B. vulgatus* strain. DNA sequences from species-specific marker genes were aligned, and then a neighbour-joining (NJ) tree was constructed based on percentage identity (PID) distance between the marker genes using Jalview. The tree is drawn to the scale bar unit (0.1) shown below the tree. The presence of the related/unrelated *B. vulgatus* strain during pregnancy was confirmed based on both WSS and StrainPhlAn (shaded colour boxes on the NJ tree). The colour boxes displayed next to the NJ tree match the label shown in figure 2.

not only the dominant strains, were transmitted to the progeny and some of these strains can amplify in the F1 to be detected in the faecal samples.

## 4. Discussion

In this study, we have used strain-tracking analysis to analyse a previously published dataset describing the faecal microbiome of mothers and infants following vaginal or Caesarean section birth. Using two different strain-tracking methods, we found the sharing of a maternal microbial strain with infants at early times (4–21 days) and at later times post-birth (infancy; 6–15 months). Importantly, we also found numerous instances where the mother and infant did and did not share faecal microbial strains. To investigate the origin of these non-shared strains, we analysed a previously described dataset of longitudinal faecal samples taken from women during pregnancy [21]. Several women had the appearance of new microbial strains at different times during pregnancy, suggesting a possible source for the observed new strains in the mother–infant pairs. However, since infant samples for these women were not available, we were unable to examine the transmission of these new strains in mothers during pregnancy. To overcome this limitation, we have used our animal models to follow the maternal strain transmission to infants more in-depth. Using an animal model surrogate to exclude the contribution of environmental microbes, gnotobiotic female mice were transplanted with five different human faecal samples and individually bred under gnotobiotic conditions. We identified both maternal-related and unrelated microbes in the F1 progeny maintained under conditions that exclude environmental microbes establishing the potential for a greater contribution of maternal strains to the infant microbiome than previously appreciated.

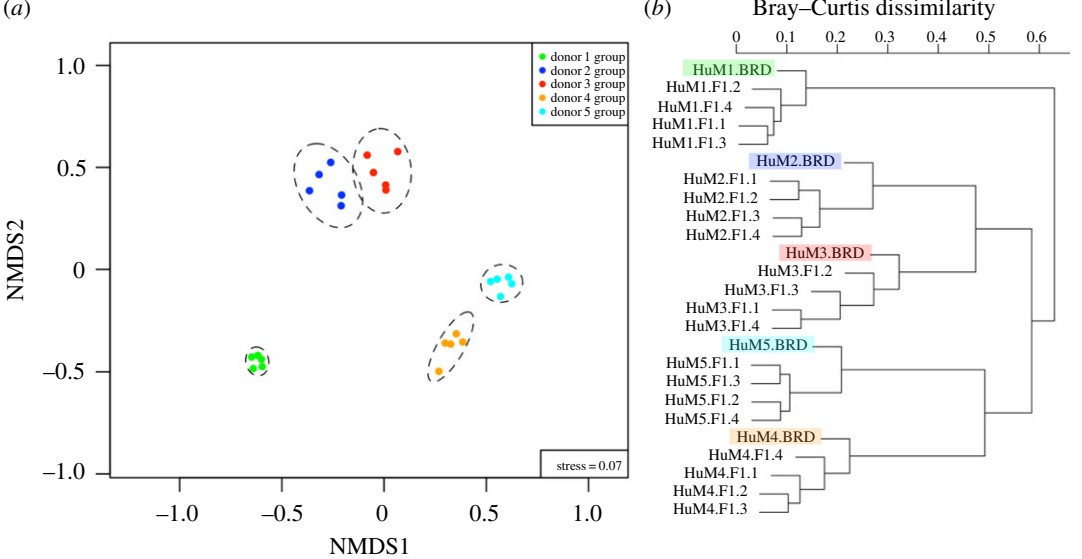

**Figure 4.** NMDS plot and cluster dendrogram of the microbial community in gnotobiotic mice and their F1 progenies post-FMT. Overall differences in the microbial community structure at the species level across all gnotobiotic mice, including dams (BRD; $n = 5$) and their F1 progenies ($n = 20$), were visualized by using (*a*) NMDS plot and (*b*) cluster dendrogram. The Bray–Curtis dissimilarities among all samples were calculated on the Hellinger-transformed 'estimated number of reads from the clade' value obtained via MetaPhlAn2 (v. 2.7.62). (*a*) The microbial community structure showed significant and distinct clusters specific to each donor group. Samples were grouped by ellipses enclosing all dots in each donor group with a confidence interval of 95% using the ordiellipse function implemented in the vegan R package. A significant difference in microbial community structure among donor groups ($R^2 =$ 0.893, *p*-value $= 1 \times 10^{-4}$) was supported by using permutational multivariate analysis of variance (PERMANOVA) with the function ADONIS in the vegan R package. (*b*) Each post-FMT dam sample (shaded colour box) was closely clustered with its F1 progenies. The cluster dendrogram was generated by using hclust function in the vegan R package.

Nearly all studies describing the complexity of the human gut microbiome analyse faecal samples. However, additional microbial diversity can be found within the gastrointestinal tract [31]. These additional microbes, the so-called dark matter of the human microbiome, are important when studies examine the stability, resistance to change and resiliency of the gut community to disturbances [32,33]. Consistent with this idea, in previous studies, we have also seen microbial strain changes in humans as a result of stress to the gut microbial community to antibiotic treatments [10]. Thus, under normal conditions, the human faecal microbiome consists of a stable dominant strain that probably represents a subset of the total microbial strains cohabitating in the gastrointestinal tract microbial community [33].

Previous studies have reported on the transmission of maternal microbes to infants as measured by the detection of maternal faecal microbial strains in the infants [12–16]. The consensus is that strains of *Bacteroides*, *Parabacteroides* and *Escherichia coli* common with the mother were transmitted to the infants during vaginal birth. However, these studies, as well as our analysis using different strain-tracking methods, also observed the individual mother and infant microbial strains that were not related. Where do these strains come from? One possibility to consider is that the origin of the 'new strains' in the infants is from the mother. In support of this, using a dataset that examined the longitudinal stability of faecal microbial strains in women during pregnancy [21], we demonstrated that seven of the nine mothers had evidence for the transient appearance of new strains of faecal microbes. Although the reason for the appearance of these new strains is unknown, one possible source could be an increase in stress during pregnancy [34]. Unfortunately, we did not have longitudinal samples from these mothers and their infants to directly test whether the new strains in the mother were transmitted to the infants. Another source for the new strains observed in the infants might be the placenta. A previous study [35] characterized a unique placental microbiome that included non-pathogenic commensal microbiota such as Firmicutes, Tenericutes, Proteobacteria, Bacteroidetes and Fusobacteria. Indeed, we found a few strains of *Bacteroides* spp. were transmitted from mother to infant following Caesarean section birth. Generally, these strains did not dominate the faecal microbiota as seen for the vaginal birth, which might indicate only a small number of microbes (possibly from the placenta) had colonized and expanded in the infant gastrointestinal tract.

**(a)**

| species | HuM1 F1–1 | HuM1 F1–2 | HuM1 F1–3 | HuM1 F1–4 |
|---|---|---|---|---|
| *Akkermansia muciniphila* (CO: 99.8) | green | green | green | green |
| *Alistipes onderdonkii* (CO: 97.4) | * | * | * | * |
| *Bacteroides massiliensis* (CO: 97.7) | * | * | * | * |
| *Bacteroides ovatus* (CO: 91.7) | * | * | * | * |
| *Bacteroides stercoris* (CO: 97.1) | * | * | * | * |
| *Bacteroides uniformis* (CO: 97.8) | red | green | green | green |
| *Bacteroides vulgatus* (CO: 95.1) | green | green | green | green |
| *Parabacteroides merdae* (CO: 98.1) | green | green | green | green |

**(b)**

| species | HuM2 F1–1 | HuM2 F1–2 | HuM2 F1–3 | HuM2 F1–4 |
|---|---|---|---|---|
| *Akkermansia muciniphila* (CO: 99.8) | red | red | green | red |
| *Alistipes putredinis* (CO: 93.9) | red | red | red | red |
| *Alistipes shahii* (CO: 92) | | red | | red |
| *Bacteroides cellulosilyticus* (CO: 98.2) | green | green | green | green |
| *Bacteroides coprocola* (CO: 67) | * | * | * | * |
| *Bacteroides ovatus* (CO: 91.7) | green | green | red | green |
| *Bacteroides* sp. 1-1-6 (CO: 97.4) | green | green | green | green |
| *Bacteroides uniformis* (CO: 97.8) | blue | blue | | |
| *Bacteroides vulgatus* (CO: 95.1) | * | red | red | red |
| *Barnesiella intestinihominis* (CO: 99.3) | | * | * | * |
| *Parabacteroides* sp. D13 (CO: 99) | * | red | red | red |

**(c)**

| species | HuM3 F1–1 | HuM3 F1–2 | HuM3 F1–3 | HuM3 F1–4 |
|---|---|---|---|---|
| *Bacteroides cellulosilyticus* (CO: 98.2) | green | green | green | green |
| *Bacteroides ovatus* (CO: 91.7) | * | red | * | * |
| *Bacteroides* sp. 1-1-6 (CO: 97.4) | * | * | * | * |
| *Bacteroides* sp. 2-1-16 (CO: 97.4) | * | * | * | * |
| *Bacteroides stercoris* (CO: 97.1) | | | | * |
| *Bacteroides uniformis* (CO: 97.8) | * | * | * | * |
| *Bacteroides vulgatus* (CO: 95.1) | | blue | | |
| *Parabacteroides* sp. D13 (CO: 99) | green | green | green | green |

**(d)**

| species | HuM4 F1–1 | HuM4 F1–2 | HuM4 F1–3 | HuM4 F1–4 |
|---|---|---|---|---|
| *Akkermansia muciniphila* (CO: 99.8) | * | * | * | * |
| *Alistipes onderdonkii* (CO: 97.4) | * | * | * | * |
| *Bacteroides cellulosilyticus* (CO: 98.2) | * | * | * | * |
| *Bacteroides ovatus* (CO: 91.7) | green | green | green | green |
| *Bacteroides stercoris* (CO: 97.1) | * | * | * | * |
| *Bacteroides uniformis* (CO: 97.8) | * | * | * | * |
| *Barnesiella intestinihominis* (CO: 99.3) | * | green | green | green |
| *Parabacteroides* sp. D13 (CO: 99) | | * | * | * |
| *Ruminococcus torques* ATCC (CO: 96.6) | red | blue | red | |

**(e)**

| species | HuM5 F1–1 | HuM5 F1–2 | HuM5 F1–3 | HuM5 F1–4 |
|---|---|---|---|---|
| *Alistipes onderdonkii* (CO: 97.4) | | | blue | |
| *Alistipes putredinis* (CO: 93.9) | * | * | * | * |
| *Bacteroides cellulosilyticus* (CO: 98.2) | red | red | red | red |
| *Bacteroides ovatus* (CO: 91.7) | | * | * | |
| *Bacteroides* sp. 1-1-6 (CO: 97.4) | * | * | * | * |
| *Bacteroides uniformis* (CO: 97.8) | red | red | red | red |
| *Parabacteroides* sp. D13 (CO: 99) | * | * | * | * |
| *Ruminococcus torques* ATCC (CO: 96.6) | | blue | blue | blue |

Legend:
- green — F1's strain was related to the breeder's strain
- green * — F1's strain was related to both breeder's and donor's strain
- blue — F1's strain was related to donor's strain, not breeder's strain
- red — no related strains observed between F1 and breeder and donor

**Figure 5.** Summarized WSS scores for the post-FMT gnotobiotic mouse dataset. Species that had a cut-off value and were able to provide a WSS score were selected from each donor group; (a) donor 1 group, (b) donor 2 group, (c) donor 3 group, (d) donor 4 group and (e) donor 5 group to compare the WSS scores between the donor and corresponding post-FMT dam (BRD) and F1 progenies. The summarized WSS scores of the species were grouped into different colour boxes (colours described in the figure labels). Each column in the table displays an individual ID and matches the label shown in electronic supplementary material, table S1. WSS scores for all identified species are provided in electronic supplementary material, tables S5 and S6.

To further address the maternal contribution to infant gut microbial strains, we used a gnotobiotic animal model surrogate with mice housed in a facility that would eliminate the possibility of new strains from the environment. Using five different dam–F1 progeny combinations, we demonstrate the transmission of related, and importantly, unrelated microbial strains to individual F1 progeny. We noted that the different dam–F1 progeny pairs had variations in strain transmission, ranging from all F1 progeny having maternal strains to variations of maternal-related and unrelated in different F1 progeny from the same litter. It is important though to acknowledge caveats with these studies. Due to coprophagy, we cannot exclude the consumption of faecal material by F1 progeny from the siblings. However, this would result in the strains between the F1 progeny being more similar. The most consistent microbial strains that were transferred from our analysis were *Bacteroides* spp. and specifically, *B. vulgatus*. Specifically, our analysis from the mother–infant dataset supports the previously described transmission of maternal strains of *B. vulgatus* to infants during vaginal birth. Although the mechanism of how maternal transfer of microbes occurs is unknown, previous studies have found that *Bacteroides* spp., and *B. vulgatus*, in particular, have a close relationship with the mammalian gut due to the presence of bacterial receptors for the gastrointestinal cells [31]. Indeed, the presence of these receptors could facilitate more efficient colonization that would be needed under conditions of low numbers of bacteria that reflect what would probably occur during maternal to infant transmission.

Finally, the results of our studies support a reconsideration of the contribution of maternal microbes to the infant enteric microbial community. The constellation of microbial strains that we detected in the infants from the mother was different in each mother–infant pair. Consequently, for individual-specific mother–infant pairs, the transmitted microbial communities could provide unique aspects of the metabolic functions found in the mother [15,36]. The differences in metabolic functions contributed by the gut microbial communities, in combination with host gene inheritance, could enhance the possibility of pathogenesis. For example, food allergies have increased in recent years and a recent genome-wide analysis found five loci involved in any food allergy including peanut allergy [37]. However, twins are

known to share microbial strains after birth and the incidence of food allergy heritability for twins is about 80% [37,38]. Moreover, germ-free mice colonized with bacteria from healthy and cow's milk allergic (CMA) infants (age about six months after birth) showed allergic responses to dietary antigens, further supporting the importance of modulating microbial communities for food allergy [39]. Whether the presence of maternal microbes enhances or protects against the development of food allergy is unknown, although it is important to note that infants delivered by Caesarean have less *Bacteroides* spp. and are known to have a greater incidence of allergies [15,40,41]. Collectively, our results provide a rationale for monitoring the microbial strain stability of mothers throughout pregnancy and the transmission of microbe strains to infants following birth. Ultimately, this analysis might provide new information for the linkage of inherited host gene diseases with dysbiosis in the gut microbial community [42,43].

Ethics. Consent form for human faecal samples was obtained as part of an ongoing IRB-approved study at the University of Alabama at Birmingham (UAB) IRB300004198. Informed consent from all donors was obtained. All mice were bred and maintained in accordance with guidelines and approval of the UAB Institutional Animal Care and Use Committee (nos. IACUC-21220 and IACUC-21645).

Data accessibility. The original sequencing datasets of the metagenomic samples used in this study were downloaded from the European Nucleotide Archive (ENA) under accession number ERP115334 for Shao *et al.* [20] and PRJNA288562 for Goltsman *et al.* [21]. The metagenomics sequencing FASTQ files from the gnotobiotic mice used in this study were deposited in the National Center for Biotechnology Information BioProject under accession number PRJNA593263. Data and relevant code for this research work are stored in GitHub: https://github.com/hkoo87/mgSNP_2 and have been archived within the Zenodo repository: https://doi.org/10.5281/zenodo.3706776.

Authors' contributions. H.K., B.C.M. and C.D.M. conceived the study and wrote the manuscript. H.K., J.A.H. and D.K.C. contributed bioinformatics analyses. M.R.C. did the NextSeq DNA sequencing. J.M.R. and E.N.B. provided reagents and reviewed the manuscript. All authors approved the final manuscript.

Competing interests. The authors declare no competing interests.

Funding. This work was supported by the University of Alabama School of Medicine (C.D.M.), R01CA194414 (E.N.B.), R03NS116559 (B.C.M.), UAB Neuro-Oncology Support Fund Award (B.C.M.) and UAB ACS-IRG O'Neal Comprehensive Cancer Center Award (B.C.M.).

Acknowledgements. We thank Shao *et al.* and Goltsman *et al.*, for publicly sharing their datasets, allowing us to conduct our study. We thank Cheaha UABgrid by UAB Information Technology's Research Computing group (ITRC) for providing the high-performance computing support necessary for bioinformatics analyses. We thank Peter Eipers, Kelly T. Goldsmith, Caitlin M. Cox and Leila Michelle for sample preparation for DNA sequencing, Kory Dees and Fraser Humphreys for gnotobiotic care and breeding and Adrienne L. Ellis for preparation of the manuscript.

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
