## [Reviewer comments · Royal Society Open Science]

Review History

RSOS-192200.R0 (Original submission)

Review form: Reviewer 1

Is the manuscript scientifically sound in its present form?

No

Are the interpretations and conclusions justified by the results?

Yes

Is the language acceptable?

Yes

Do you have any ethical concerns with this paper?

No

Have you any concerns about statistical analyses in this paper?

No

Recommendation?

Major revision is needed (please make suggestions in comments)

Comments to the Author(s)

2020 Koo Review

Summary - Understanding the maternal and environmental origins of the infant gastrointestinal (GI) is an important field of study as short- and long-term health of the infant is associated with the infant's GI microbiome. In this manuscript, the authors utilize two publicly available data sets to investigate the transfer of specific strains between mother and infant. The authors also report on a gnotobiotic mouse study that examines transfer of strains between mother and F1 progeny. Key conclusions are that there is an individualized transfer of strains as not all strains are equally transferred between mothers and their infants (this was also reflected in the gnotobiotic mouse study), there are also many differences between the strains found in feces of a mother and her infant, and many strains are related during pregnancy.

Introduction

Line 61-64: Incomplete sentence

Line 64-65: "Next-generation Sequencing" does not need have the capital letters.

Materials and Methods

Line 91: When were the feces samples collected from the mothers?

Line 94 or Table S1: Where is information for the 6 mother-infants from the caesarean part of the analysis?

Line 125: What happened to Donor 5 feces? Also not reported in Table S1.

Table S1 - There are multiple samples from the female gnotobiotic mice post FMT with Donor 4 and Donor 5. What is the difference between these samples and were they all used when making the comparisons? It does appear the samples were merged.

Line 143: Once again, when was the mother's sample obtained?

Line 144: Was the mother's sample only compared to her infant's samples? Please clarify as when I first read this it says "any available infant samples". Upon further reading, I am guessing each mother's sample was only compared with her infant's samples.

Line 168: The authors are commended for providing the resource/link that allows for reproduction of the analyses.

Line 169, Table S2 has more than 20. Please change title to reflect the number of species reported in table. And are these the top 41 most abundant species?

Line 176-177: I do not think you need to list Tables S3-S6 here. Instead they should be listed where they are discussed in the Results.

Line 191: This sentence seems to be missing something. Perhaps a "which" before "included 'estimated number of reads...'"?

Results

Line 204 - 209: These sentences seem out of place here and are more discussion than results.

Line 209- 212: Redundant with information in Materials and Methods.

Line 212: Change in font (actually there are changes in the font in several places across the manuscript).

Table S3: If the comparison was only conducted between each mother and her infant, please change title to reflect that, e.g. "between each mother vs her infant". If this is not the case, then more detail is needed to describe that this reflects a mother vs all infants.

Table S3: Row 2 – should this be "Fecal sample from infant"?

Table S3: Includes 23 species.

Table S3: What does a "NA" mean for the CO? What is the cutoff value used when CO is NA?

Line 213 – 215: This sentence that describes the top 20 in Table S2 seems like it should be reported first. And then move to reporting Table S3.

Line 216-217: Once again, first sentence of paragraph seems redundant.

Figure 1: When comparing Table S3 and Figure 1, there seem to be several inconsistencies. For example, For *Bacteroides* sp 1-1-6, shouldn't the boxes for B01948, B02273, and C01917 also include blue? For *Bacteroides* 2 1 16, shouldn't C01994 also include blue? Perhaps you can separate the box into triangles to display two colors.

Line 220: The authors report 13 of 16 pairs had *B. vulgatus*. However, only 9 of 13 share related strains (as defined by > CO threshold). Perhaps the authors can further clarify that for the reader as well since the focus of the paper seems to be on the relatedness of the species. Along with that, I think having just a bit more information either in the introduction or in the materials and methods about the WSS method and the setting of the CO threshold would be helpful to the reader so they do not have to go back and find this information through reading of the references.

Line 222 – 224: The authors make special note of *Akkermansia muciniphila*, *Barnesiella intestinihominis*, and *Faecalibacterium prausnitzii*. These seem odd to point out. And if pointing out those that are not related, why not also include *Eubacterium hallii* and *Ruminococcus*?

Line 224-225: This sentence seems very much like discussion.

Line 227-230: If using the red font as indicator of being related or not, I count 2 related between D4-21 and not at infancy, 2 that had unrelated at between D4-21 but related at infancy, 5 that had related between D4-21 and at infancy and 4 unrelated at any time point. Please address the discrepancy or where I am perhaps mistaken.

Line 231: I interpret Figure 1 as showing that C01917 mother-infant pair only shared 5 of the bacterial strains; not all detected microbial strains.

Line 236: Which 6 cases agree? Please detail for the reader.

Line 239-242: This is not described in Materials and Methods. Please add this.

Table S8: Should row 2 be "Fecal sample from infant"?

Table S8: Why are there only 9 species reported on?

Line 248: The authors should also discuss the limitation that only 6 mother-infant pairs were used in the caesarean analyses.

Line 251-256: Once again, these sentences seem to more appropriate for the Discussion and not in the results section.

Line 257-258: Not sure how this supports the transfer of microbes to the infants since it is only comparing strains across pregnancy and does not then compare to what is found in her infant. It is a worthwhile description as to strains that remain stable or that are more transient in nature but I do not quite see the connection that demonstrates subsequent transfer to infants.

Figure 3: Individual Pre4 has a green box for *Bacteroides* sp. 2-1-16 indicating no strain changes during any time points of pregnancy. However, this is misleading as there were no samples available for the other two time points (both are marked as “NA” in Table S4). Therefore, I would say it is unknown whether there were any strain changes or not.

Perhaps I am not understanding this figure but I would also disagree with the green box for Term6 in the *Bacteroides stercoris* row. If there were no strain changes during any time points of pregnancy, wouldn't there be red numbers in Table S4 for this individual for that strain?

I did not go through every box but please address these inconsistencies and/or clarify in the text so readers don't perhaps make the same misinterpretation that I am making.

Line 261: Why is this important?

Line 264: Please provide which 5 cases agreed.

Line 269-271: Once again, this sentence seems more like discussion, at least from looking at other articles in the journal; but perhaps I am mistaken.

Line 271-278: This seems redundant to what is in the Materials and Methods and / or any detail included here should be added to the Materials and Methods section.

Line 277: “was” should be “were”

Line 287: Results from the comparisons found in Tables S5-S6 are lacking. Figure 6 does do a nice job of pulling the results together, but perhaps more of the results can be presented or detailed in this section as well.

Also please direct the readers to these supplemental tables in this section.

Line 288: would recommend amending the sentence to read “maternal transmission of bacteria to the F1s...” or specify transmission of what.

Line 289: Once again, in several places in the manuscript there is a font change.

Line 298: And also caesarean birth, correct?

Line 298-301: I find this sentence somewhat misleading as very few strains were observed in early life of the infant and maintained at infancy, at least according to Table S3 and Figure 1.

Line 304-306: How does appearance of strains during pregnancy suggest changes in observed strains seen postpartum for the mother and during early life of the infant?

Line 326-332: I think the authors are missing a critical component, the diet of mother and infant, that most likely plays a key role in structuring fecal bacterial communities. This should be discussed.

Line 338: Should “from caesarean birth” instead be “following caesarean birth”?

Line 348: This reviewer appreciates the authors acknowledging caveats of the study. In particular, it is noted that sampling of the F1 progeny's feces was done at time of weaning. Please remind the readers as to when the dams were sampled as well. I think there would have been about a 7-day difference between the two sampling, correct?

References

Line 426: Segata reference is missing additional information

Segata N. 2018. On the road to strain-resolved comparative metagenomics. *mSystems* 3:e00190-17.

Line 440: "microbial" is misspelled.

Figure legends

Line 580: What was the cutoff for "abundant species"?

Review form: Reviewer 2 (Maria Gloria Dominguez-Bello)

Is the manuscript scientifically sound in its present form?

Yes

Are the interpretations and conclusions justified by the results?

Yes

Is the language acceptable?

Yes

Do you have any ethical concerns with this paper?

No

Have you any concerns about statistical analyses in this paper?

No

Recommendation?

Accept as is

Comments to the Author(s)

The analyses are elegantly done, and although the results are not groundbreaking, they elegantly confirm that engraftment in new niches brings microbes that are not observed in the source, because they were in very low load (rare).

My only observation is that the text of their manuscript has inconsistent fonts. I am sure this is fixable by the journal editors.

Decision letter (RSOS-192200.R0)

06-Feb-2020

Dear Ms Koo,

The editors assigned to your paper ("An Individualized Mosaic of Maternal Microbial Strains are Transmitted to the Infant Gut Microbial Community") have now received comments from reviewers. We would like you to revise your paper in accordance with the referee and Associate Editor suggestions which can be found below (not including confidential reports to the Editor). Please note this decision does not guarantee eventual acceptance.

Please submit a copy of your revised paper before 29-Feb-2020. Please note that the revision

deadline will expire at 00.00am on this date. If we do not hear from you within this time then it will be assumed that the paper has been withdrawn. In exceptional circumstances, extensions may be possible if agreed with the Editorial Office in advance. We do not allow multiple rounds of revision so we urge you to make every effort to fully address all of the comments at this stage. If deemed necessary by the Editors, your manuscript will be sent back to one or more of the original reviewers for assessment. If the original reviewers are not available, we may invite new reviewers.

- Data accessibility

If you wish to submit your supporting data or code to Dryad (<http://datadryad.org/>), or modify your current submission to dryad, please use the following link:
<http://datadryad.org/submit?journalID=RSOS&manu=RSOS-192200>

- Competing interests

- Authors' contributions

- Acknowledgements

- Funding statement

on behalf of Dr Ulas Tezel (Associate Editor) and Kevin Padian (Subject Editor)
openscience@royalsociety.org

Editor comments:

As you will see we have divergent reviews of your manuscript. One reviewer still has many comments to address. Please answer them assiduously. If you need more time, please notify the editorial office. Best wishes.

Reviewers' Comments to Author:

Reviewer: 1

Comments to the Author(s)

2020 Koo Review

Summary - Understanding the maternal and environmental origins of the infant gastrointestinal (GI) is an important field of study as short- and long-term health of the infant is associated with the infant's GI microbiome. In this manuscript, the authors utilize two publicly available data sets to investigate the transfer of specific strains between mother and infant. The authors also report on a gnotobiotic mouse study that examines transfer of strains between mother and F1 progeny. Key conclusions are that there is an individualized transfer of strains as not all strains are equally transferred between mothers and their infants (this was also reflected in the gnotobiotic mouse study), there are also many differences between the strains found in feces of a mother and her infant, and many strains are related during pregnancy.

Introduction

Line 61-64: Incomplete sentence

Line 64-65: "Next-generation Sequencing" does not need have the capital letters.

Materials and Methods

Line 91: When were the feces samples collected from the mothers?

Line 94 or Table S1: Where is information for the 6 mother-infants from the caesarean part of the analysis?

Line 125: What happened to Donor 5 feces? Also not reported in Table S1.

Table S1 - There are multiple samples from the female gnotobiotic mice post FMT with Donor 4 and Donor 5. What is the difference between these samples and were they all used when making the comparisons? It does appear the samples were merged.

Line 143: Once again, when was the mother's sample obtained?

Line 144: Was the mother's sample only compared to her infant's samples? Please clarify as when I first read this it says "any available infant samples". Upon further reading, I am guessing each mother's sample was only compared with her infant's samples.

Line 168: The authors are commended for providing the resource/link that allows for reproduction of the analyses.

Line 169, Table S2 has more than 20. Please change title to reflect the number of species reported in table. And are these the top 41 most abundant species?

Line 176-177: I do not think you need to list Tables S3-S6 here. Instead they should be listed where they are discussed in the Results.

Line 191: This sentence seems to be missing something. Perhaps a "which" before "included 'estimated number of reads...'"?

Results

Line 204 - 209: These sentences seem out of place here and are more discussion than results.

Line 209- 212: Redundant with information in Materials and Methods.

Line 212: Change in font (actually there are changes in the font in several places across the manuscript).

Table S3: If the comparison was only conducted between each mother and her infant, please change title to reflect that, e.g. "between each mother vs her infant". If this is not the case, then more detail is needed to describe that this reflects a mother vs all infants.

Table S3: Row 2 - should this be "Fecal sample from infant"?

Table S3: Includes 23 species.

Table S3: What does a "NA" mean for the CO? What is the cutoff value used when CO is NA?

Line 213 - 215: This sentence that describes the top 20 in Table S2 seems like it should be reported first. And then move to reporting Table S3.

Line 216-217: Once again, first sentence of paragraph seems redundant.

Figure 1: When comparing Table S3 and Figure 1, there seem to be several inconsistencies. For example, For *Bacteroides* sp 1-1-6, shouldn't the boxes for B01948, B02273, and C01917 also

include blue? For *Bacteroides* 2-1-16, shouldn't C01994 also include blue? Perhaps you can separate the box into triangles to display two colors.

Line 220: The authors report 13 of 16 pairs had *B. vulgatus*. However, only 9 of 13 share related strains (as defined by > CO threshold). Perhaps the authors can further clarify that for the reader as well since the focus of the paper seems to be on the relatedness of the species. Along with that, I think having just a bit more information either in the introduction or in the materials and methods about the WSS method and the setting of the CO threshold would be helpful to the reader so they do not have to go back and find this information through reading of the references.

Line 222 – 224: The authors make special note of *Akkermansia muciniphila*, *Barnesiella intestinihominis*, and *Faecalibacterium prausnitzii*. These seem odd to point out. And if pointing out those that are not related, why not also include *Eubacterium hallii* and *Ruminococcus*?

Line 224-225: This sentence seems very much like discussion.

Line 227-230: If using the red font as indicator of being related or not, I count 2 related between D4-21 and not at infancy, 2 that had unrelated at between D4-21 but related at infancy, 5 that had related between D4-21 and at infancy and 4 unrelated at any time point. Please address the discrepancy or where I am perhaps mistaken.

Line 231: I interpret Figure 1 as showing that C01917 mother-infant pair only shared 5 of the bacterial strains; not all detected microbial strains.

Line 236: Which 6 cases agree? Please detail for the reader.

Line 239-242: This is not described in Materials and Methods. Please add this.

Table S8: Should row 2 be “Fecal sample from infant”?

Table S8: Why are there only 9 species reported on?

Line 248: The authors should also discuss the limitation that only 6 mother-infant pairs were used in the caesarean analyses.

Line 251-256: Once again, these sentences seem to more appropriate for the Discussion and not in the results section.

Line 257-258: Not sure how this supports the transfer of microbes to the infants since it is only comparing strains across pregnancy and does not then compare to what is found in her infant. It is a worthwhile description as to strains that remain stable or that are more transient in nature but I do not quite see the connection that demonstrates subsequent transfer to infants.

Figure 3: Individual Pre4 has a green box for *Bacteroides* sp. 2-1-16 indicating no strain changes during any time points of pregnancy. However, this is misleading as there were no samples available for the other two time points (both are marked as “NA” in Table S4). Therefore, I would say it is unknown whether there were any strain changes or not.

Perhaps I am not understanding this figure but I would also disagree with the green box for Term6 in the *Bacteroides stercoris* row. If there were no strain changes during any time points of pregnancy, wouldn't there be red numbers in Table S4 for this individual for that strain? I did not go through every box but please address these inconsistencies and/or clarify in the text so readers don't perhaps make the same misinterpretation that I am making.

Line 261: Why is this important?

Line 264: Please provide which 5 cases agreed.

Line 269-271: Once again, this sentence seems more like discussion, at least from looking at other articles in the journal; but perhaps I am mistaken.

Line 271-278: This seems redundant to what is in the Materials and Methods and / or any detail included here should be added to the Materials and Methods section.

Line 277: "was" should be "were"

Line 287: Results from the comparisons found in Tables S5-S6 are lacking. Figure 6 does do a nice job of pulling the results together, but perhaps more of the results can be presented or detailed in this section as well.

Also please direct the readers to these supplemental tables in this section.

Line 288: would recommend amending the sentence to read "maternal transmission of bacteria to the F1s..." or specify transmission of what.

Line 289: Once again, in several places in the manuscript there is a font change.

Line 298: And also caesarean birth, correct?

Line 298-301: I find this sentence somewhat misleading as very few strains were observed in early life of the infant and maintained at infancy, at least according to Table S3 and Figure 1.

Line 304-306: How does appearance of strains during pregnancy suggest changes in observed strains seen postpartum for the mother and during early life of the infant?

Line 326-332: I think the authors are missing a critical component, the diet of mother and infant, that most likely plays a key role in structuring fecal bacterial communities. This should be discussed.

Line 338: Should "from caesarean birth" instead be "following caesarean birth"?

Line 348: This reviewer appreciates the authors acknowledging caveats of the study. In particular, it is noted that sampling of the F1 progeny's feces was done at time of weaning. Please remind the readers as to when the dams were sampled as well. I think there would have been about a 7-day difference between the two sampling, correct?

References

Line 426: Segata reference is missing additional information

Segata N. 2018. On the road to strain-resolved comparative metagenomics. *mSystems* 3:e00190-17.

Line 440: "microbial" is misspelled.

Figure legends

Line 580: What was the cutoff for "abundant species"?

Reviewer: 2

Comments to the Author(s)

The analyses are elegantly done, and although the results are not groundbreaking, they elegantly confirm that engraftment in new niches brings microbes that are not observed in the source, because they were in very low load (rare).

My only observation is that the text of their manuscript has inconsistent fonts. I am sure this is fixable by the journal editors.

Author's Response to Decision Letter for (RSOS-192200.R0)

See Appendix A.

Decision letter (RSOS-192200.R1)

24-Mar-2020

Dear Dr Koo,

It is a pleasure to accept your manuscript entitled "An Individualized Mosaic of Maternal Microbial Strains are Transmitted to the Infant Gut Microbial Community" in its current form for publication in Royal Society Open Science.

Kind regards,

on behalf of Dr Ulas Tezel (Associate Editor) and Kevin Padian (Subject Editor)
openscience@royalsociety.org

Appendix A

March 13, 2020

Andrew Dunn
Royal Society Open Science Editorial Office
Royal Society Open Science
openscience@royalsociety.org

Dear Dr. Dunn,

We are returning the revised manuscript “An Individualized Mosaic of Maternal Microbial Strains are Transmitted to the Infant Gut Microbial Community” for consideration in Open Science. As per instruction, we have included a marked up version of the manuscript (changes are noted in red fonts), a clean revised version of the manuscript, and response to referees with notations to the line changes that correspond to both versions of the manuscript.

Note that as requested by RSOS editorial office that we have provided BioProject accession number (PRJNA593263) (now publicly released) and provided the data accessibility statement as “Data and relevant code for this research work are stored in GitHub: https://github.com/hkoo87/mgSNP_2 and have been archived within the Zenodo repository: <https://doi.org/10.5281/zenodo.3706776>”.

We have incorporated all of the Reviewer 1 comments (52 in total) although we note that in several instances there appeared to be a duplication of the comments. In the revised manuscript, we have not changed the overall emphasis or conclusions and, in our opinion, the modifications suggested by Reviewer 1 have improved the flow of the manuscript.

Specifically, we have revised the original Figure 1 to include Figure 1A and Figure 1B to clarify our analysis. The text in the Results for this figure has been modified to more clearly point out the related and unrelated microbial strains in the mother-infant pairs at different times post-birth. Figure 2 in the original version has now been placed in Supplement Figure 1 (S1 Figure). Figures 2 and 3 (Figures 3 and 4 in the original manuscript) have been modified to more clearly represent related and unrelated microbial strains during days of pregnancy. We have also modified Figure 5 (Figure 6 in the original manuscript) to present our data from the individual mother (dam) and her F1’s. In addition, we have now included strain-tracking analysis of Donor 5 fecal samples that were missing in the original version of the paper. Lastly, we have accordingly revised the Supplementary Tables (S1 to S8) to clearly support the revised figures. As suggested by the RSOS editorial office, we have provided a ESM Legends file which includes titles and short descriptions for each supplementary table and figure.

We believe with the incorporation of the modifications the manuscript has been improved and will be of interest to the readers of Open Science.

Best regards,

Hyunmin Koo, PhD (corresponding author)

Casey D. Morrow, PhD (corresponding author)

REVIEWER 1

Summary – Understanding the maternal and environmental origins of the infant gastrointestinal (GI) is an important field of study as short- and long-term health of the infant is associated with the infant’s GI microbiome. In this manuscript, the authors utilize two publicly available data sets to investigate the transfer of specific strains between mother and infant. The authors also report on a gnotobiotic mouse study that examines transfer of strains between mother and F1 progeny. Key conclusions are that there is an individualized transfer of strains as not all strains are equally transferred between mothers and their infants (this was also reflected in the gnotobiotic mouse study), there are also many differences between the strains found in feces of a mother and her infant, and many strains are related during pregnancy.

Overall response:

We thank Reviewer 1 for the very thorough review of the manuscript. We have made every effort to incorporate all of the reviewer’s suggestions into the revised manuscript. In doing so, we believe the manuscript has been substantially improved.

In particular, we have revised the original Figure 1 to include Figure 1A and Figure 1B to clarify our analysis of the Zhao et al., data set. The text in the Results for this figure has been modified to more clearly point out the related and unrelated microbial strains in the mother-infant pairs at different times post-birth. Figure 2 in the original version has now been placed in Supplementary Figure 1 (S1 Figure). Figures 2 and 3 (Figures 3 and 4 in the original manuscript) have also been modified to more clearly represent related and unrelated microbial strains during days of pregnancy. In our opinion, these modifications have improved the flow of the manuscript.

In the revised version, we have also modified Figure 5 (Figure 6 in the original manuscript) to present our data from the individual dam and her F1’s. In addition, we have now included strain-tracking analysis of Donor 5 fecal samples that were missing in the original version of the paper. We have accordingly reviewed all Supplementary Tables (S1 to S8) to clearly support the revised figures. Please note that the number of Supplementary Tables (S4 to S8) are changed and correspond to the revised text.

Comment 1: Line 61-64: Incomplete sentence

Response 1: As suggested by the reviewer, we have corrected sentences to the revised manuscript (Line numbers: 61-64).

Comment 2: Line 64-65: “Next-generation Sequencing” does not need have the capital letters.

Response 2: We have changed the word from “Next-generation Sequencing” to “next-generation sequencing” in the revised manuscript (Line numbers: 64).

Comment 3: Line 91: When were the feces samples collected from the mothers?

Response 3: Please note that exact sample collection times for the mothers were not reported by Shao et al. However, we have included sample collection information for the mothers in the revised manuscript (Line numbers: 96-97) based on the information that Shao et al. provided in their study.

Comment 4: Line 94 or Table S1: Where is information for the 6 mother-infants from the caesarean part of the analysis?

Response 4: Thank you for pointing it out. We have included the caesarean sample information ($n=6$) in the revised manuscript (Line numbers: 99-101) as well as Tables S1(A) and S2(A).

Comment 5: Line 125: What happened to Donor 5 feces? Also not reported in Table S1.

Response 5: Thank you for pointing it out. We have included Donor 5 information in the revised manuscript (Line number: 113, 128, 604-607) as well as Table S1(C). We have also revised the Tables S2(C), S6, S7, S8 and Figure 5 by including Donor 5 information.

Comment 6: Table S1 – There are multiple samples from the female gnotobiotic mice post FMT with Donor 4 and Donor 5. What is the difference between these samples and were they all used when making the comparisons? It does appear the samples were merged.

Response 6: Thank you for pointing it out. In the revised manuscript, we have included new information for only one female gnotobiotic mouse and her corresponding infants for Donor 3, 4, and 5. Accordingly, we have revised Tables S1(C), S2(C), S6, S7, and Figure 5 (Line numbers: 128, 604-607). We also changed the word “breeder” to “dam” throughout the manuscript.

Comment 7: Line 143: Once again, when was the mother’s sample obtained?

Response 7: As we already mentioned in Response 3, we have included sample collection information for the mothers in the revised manuscript (Line numbers: 96-97).

Comment 8: Line 144: Was the mother’s sample only compared to her infant’s samples? Please clarify as when I first read this it says “any available infant samples”. Upon further reading, I am guessing each mother’s sample was only compared with her infant’s samples.

Response 8: Please note that each mother’s sample was only compared with her infant’s samples. We have clarified this sentence in the revised manuscript (Line numbers: 145-146).

Comment 9: Line 168: The authors are commended for providing the resource/link that allows for reproduction of the analyses.

Response 9: Thank you. Please note that the metagenomics sequencing FASTQ files from the gnotobiotic mice used in this study were deposited in NCBI under accession number PRJNA593263.

Comment 10: Line 169, Table S2 has more than 20. Please change title to reflect the number of species reported in table. And are these the top 41 most abundant species?

Response 10: Please note that the value in the Table S2 represents the number of sample pairs for each species after filtering any low sequence coverage (<30%) and low sequence depth (<3.5) samples against their given reference sequences via WSS. We have already reported these parameters in the Materials and Methods section of the previous version of the manuscript.

However, we understand the reviewer's point regarding abundant species and the top 20 species. Thus, instead of selecting the top 20 species across all three different data sets, we have selected all available species that had a cut-off value and were able to provide WSS scores from each data set. We have accordingly included this information in Table S2 along with the title and legends, Figures 1, 2, and 5, and in the text of the revised manuscript (Line numbers: 170-171, 176, 206-210, 262-265, 288-291, 554, 565, 604, 620-626).

Comment 11: Line 176-177: I do not think you need to list Tables S3-S6 here. Instead they should be listed where they are discussed in the Results.

Response 11: As suggested by the reviewer, we have revised the sentence by removing the listed Tables S3-S6 (Line number: 178). Instead, we have included Tables S3-S8 information in the Results section of the revised manuscript (Line numbers: 212-214, 246-248, 262-265, 291-294).

Comment 12: Line 191: This sentence seems to be missing something. Perhaps a "which" before "included 'estimated number of reads...'"?

Response 12: As suggested by the reviewer, we have modified the sentence by including "which" before "included 'estimated number of reads..". (Line number: 190)

Comment 13: Line 204 – 209: These sentences seem out of place here and are more discussion than results.

Response 13: We agree and have modified this section in the revised manuscript (Line numbers: 204-206).

Comment 14: Line 209- 212: Redundant with information in Materials and Methods.

Response 14: As suggested by the reviewer, we have removed the redundant sentences from the manuscript (Line numbers: 204-206).

Comment 15: Line 212: Change in font (actually there are changes in the font in several places across the manuscript).

Response 15: Thank you for pointing it out. We have made every effort to maintain the same font in the revised version of the manuscript.

Comment 16: Table S3: If the comparison was only conducted between each mother and her infant, please change title to reflect that, e.g. “between each mother vs her infant”. If this is not the case, then more detail is needed to describe that this reflects a mother vs all infants.

Response 16: As suggested by the reviewer, we have revised the legend for Table S3 by changing the sentence to “between each mother’s sample and her infant’s samples (4-, 7-, 21-day, and infancy).” (Line number: 629-630)

Comment 17: Table S3: Row 2 – should this be “Fecal sample from infant”?

Response 17: Please note that we have previously written “Fecal sample from mother” in the Row2 of Table S3 to represent the Row2 was used to compare with each column of C, D, E, and F (infant samples). However, we understand the reviewer’s point, thus we have revised the Row 2 along with Columns C, D, E, and F to clarify sample information.

Comment 18: Table S3: Includes 23 species.

Response 18: Please note that Table S3 has been revised and now includes a total of 23 species for vaginal birth samples. After filtering species that did not have a cut-off value, 17 species were able to be used for further comparison. The result for the caesarean section birth samples can be found in Table S4. The revised Table S2(A) is now included for both Tables S3 and S4 information. We have clarified this information in the revised manuscript (Line number: 206-210).

Comment 19: Table S3: What does a “NA” mean for the CO? What is the cutoff value used when CO is NA?

Response 19: The “CO: NA” indicates that there was no cut-off (CO) value assigned to the species. Since there was no value assigned to the species, we have excluded these species to distinguish related sample pairs. Note that the cut-off values were determined from our previous study (Kumar et al., npj Biofilms and Microbiome) using the HMP data set base as discussed in the paper.

To avoid confusion, any WSS scores that were observed for the “species (CO: NA)” are colored in black in Tables S3, S4, S5, S6, and S7. We have also revised the

legend for Tables S3, S4, S5, S6, and S7 for clarity (Line number: 632-633, 639-640, 646-647, 653-654, 659-660).

Comment 20: Line 213 – 215: This sentence that describes the top 20 in Table S2 seems like it should be reported first. And then move to reporting Table S3.

Response 20: As suggested by the reviewer, we have reported the Table S2 first and then reported Table S3 (Line numbers: 206-210, 212-214).

Comment 21: Line 216-217: Once again, first sentence of paragraph seems redundant.

Response 21: As suggested by the reviewer, we have modified this section and eliminated the redundant sentence from the manuscript (Line number: 211).

Comment 22: Figure 1: When comparing Table S3 and Figure 1, there seem to be several inconsistencies. For example, For *Bacteroides* sp 1-1-6, shouldn't the boxes for B01948, B02273, and C01917 also include blue? For *Bacteroides* 2 1 16, shouldn't C01994 also include blue? Perhaps you can separate the box into triangles to display two colors.

Response 22: Thank you for pointing it out. As we already mentioned in Response 10, we have selected all available species and then revised Figure 1. To clarify the sample comparisons, we have provided Figure 1(A) showing pairwise comparisons conducted between each mother's sample and her infant's infancy (6-15 months, data from Zhao et al.,) sample following vaginal birth.

Additionally, we have provided Figure 1(B) for clarification that shows all pairwise comparisons conducted between each mother's sample and her infant's other time points samples (4, 7, and 21 day).

We thank the reviewer for the comments and believe the revised Figure 1(A) and 1(B) clarifies the analysis and satisfies the reviewer's comments.

Comment 23: Line 220: The authors report 13 of 16 pairs had *B. vulgatus*. However, only 9 of 13 share related strains (as defined by > CO threshold). Perhaps the authors can further clarify that for the reader as well since the focus of the paper seems to be on the relatedness of the species. Along with that, I think having just a bit more information either in the introduction or in the materials and methods about the WSS method and the setting of the CO threshold would be helpful to the reader so they do not have to go back and find this information through reading of the references.

Response 23: Actually, the focus of the analysis in this section is to identify the related AND unrelated mother to infant strains that form the foundation for the additional analysis of pregnant mothers and gnotobiotic animals.

As suggested by the reviewer, we have clarified the sentences by including detailed results observed from the revised Figure 1 as well as Table S3 (Line numbers 214-225, 228-237). Additionally, we have revised the Introduction by including detailed information regarding WSS and cut-off value (Line numbers: 71-74).

Comment 24: Line 222 – 224: The authors make special note of Akkermansia muciniphila, Barnesiella intestinihominis, and Faecalibacterium prausnitzii. These seem odd to point out. And if pointing out those that are not related, why not also include Eubacterium hallii and Ruminococcus?

Response 24: Thank you for pointing it out. As suggested by the reviewer, we have revised the manuscript by including only those species that did not have a related strain at all examined time points that were available in the data sets (Line numbers: 220-222).

Comment 25: Line 224-225: This sentence seems very much like discussion.

Response 25: As suggested by the reviewer, we have moved this sentence to the Discussion section of the revised manuscript (Line number: 379-381).

Comment 26: Line 227-230: If using the red font as indicator of being related or not, I count 2 related between D4-21 and not at infancy, 2 that had unrelated at between D4-21 but related at infancy, 5 that had related between D4-21 and at infancy and 4 unrelated at any time point. Please address the discrepancy or where I am perhaps mistaken.

Response 26: As we already mentioned in Response 23, we have now revised Figure 1 as well as Tables S2 and S3 to clarify pairwise comparisons. We have accordingly included this information in the main text (Line numbers: 214-225, 228-237)

Comment 27: Line 231: I interpret Figure 1 as showing that C01917 mother-infant pair only shared 5 of the bacterial strains; not all detected microbial strains.

Response 27: Thank you for pointing it out. As we already mentioned in Response 26, we have revised the manuscript accordingly based on the revised Figure 1 as well as the Table S3 (Line numbers: 214-225, 228-237).

We have also included information in the Abstract section of the revised manuscript to reflect these changes (Line numbers: 46-47).

Comment 28: Line 236: Which 6 cases agree? Please detail for the reader.

Response 28: As suggested by the reviewer, we have revised the manuscript accordingly by including detailed information for comparison analysis between WSS and StrainPhlAn (Line numbers: 240-241).

Please also note that we have changed “Figure 2” to “S1 Figure” since we changed Figure 1 as suggested by the reviewer.

Comment 29: Line 239-242: This is not described in Materials and Methods. Please add this.

Response 29: As we mentioned in Response 4, we have included the caesarean sample information in the revised manuscript (Line numbers: 99-101) as well as Tables S1(A) and S2(A).

Comment 30: Table S8: Should row 2 be “Fecal sample from infant”?

Response 30: As we mentioned in Response 17, we have revised Row 2 in Table S4 along with Columns C, D, E, and F to clarify sample information.

Comment 31: Table S8: Why are there only 9 species reported on?

Response 31: Please note that the species that we reported in Table S4, as well as Tables S2, S3, S5, S6, and S7 were selected based on sequence coverage and depth against reference sequences. Moreover, based on metadata provided by Shao et al., there were only two samples (B00075, BBS0070) that had a normal Bacteroides profile and the remaining samples had low Bacteroides profile. We believe this may lower the number of species that can be compared for the caesarean section birth data set.

Comment 32: Line 248: The authors should also discuss the limitation that only 6 mother-infant pairs were used in the caesarean analyses.

Response 32: As suggested by the reviewer, we have included a caveat of sample size for the caesarean section birth samples in the revised manuscript (Line numbers: 255-256).

Comment 33: Line 251-256: Once again, these sentences seem to more appropriate for the Discussion and not in the results section.

Response 33: We have modified this section in the revised manuscript (Line numbers: 259-262).

Comment 34: Line 257-258: Not sure how this supports the transfer of microbes to the infants since it is only comparing strains across pregnancy and does not then compare to what is found in her infant. It is a worthwhile description as to strains that remain stable or that are more transient in nature but I do not quite see the connection that demonstrates subsequent transfer to infants.

Response 34: We believe our Response 46 is related to the reviewer’s concern.

Comment 35: Figure 3: Individual Pre4 has a green box for Bacteroides sp. 2-1-16 indicating no strain changes during any time points of pregnancy. However, this is misleading as there were no samples available for the other two time points (both are marked as “NA” in Table S4). Therefore, I would say it is unknown whether there were any strain changes or not.

Perhaps I am not understanding this figure but I would also disagree with the green box for Term6 in the Bacteroides stercoris row. If there were no strain changes during any time points of pregnancy, wouldn't there be red numbers in Table S4 for this individual for that strain?

I did not go through every box but please address these inconsistencies and/or clarify in the text so readers don't perhaps make the same misinterpretation that I am making.

Response 35: Thank you for pointing it out. To avoid any inconsistency matches that can be caused by an unknown factor such as “NS = No Score”, we have thoroughly revised Figure 2 as well as Table S5.

As we already mentioned in the previous version of the manuscript, gestational days for 9 women were different. Out of 9 women, 3 women (Pre 1, Pre 2, and Pre 4) gave birth prematurely, therefore a total number of samples for these women are different as compared to the remaining 6 women who gave birth after 37 weeks. This was why only three women had “NA” values in Table S5. Thus, we still believe no strain changes observed for Pre 4 sample during pregnancy, since her gestational day ended before 150 days. We believe the revised Figure 2 will satisfy the reviewer.

Comment 36: Line 261: Why is this important?

Response 36: We have revised this sentence in the revised version of the manuscript (Line number: 268).

Comment 37: Line 264: Please provide which 5 cases agreed.

Response 37: As suggested by the reviewer, we have included a list of 5 cases that we showed in Figure 3 (Line number: 271-272).

Comment 38: Line 269-271: Once again, this sentence seems more like discussion, at least from looking at other articles in the journal; but perhaps I am mistaken.

Response 38: We have modified this section in the revised manuscript (Line numbers: 276-278).

Comment 39: Line 271-278: This seems redundant to what is in the Materials and Methods and / or any detail included here should be added to the Materials and Methods section.

Response 39: As suggested by the reviewer, we have removed redundant sentences (Line numbers: 278-280) and revised the Materials and Methods section by including any missed information (Line numbers: 120-124).

Comment 40: Line 277: “was” should be “were”

Response 40: This sentence has removed as suggested by the reviewer in Comment 39.

Comment 41: Line 287: Results from the comparisons found in Tables S5-S6 are lacking. Figure 6 does do a nice job of pulling the results together, but perhaps more of the results can be presented or detailed in this section as well.

Also please direct the readers to these supplemental tables in this section.

Response 41: As suggested by the reviewer, we have included detailed results for Figure 5 as well as Tables S6-S7 in the revised manuscript (Line numbers 295-306, 311-319).

Comment 42: Line 288: would recommend amending the sentence to read “maternal transmission of bacteria to the F1s...” or specify transmission of what.

Response 42: As suggested by the reviewer, we have changed the sentence from “maternal transmission to the F1s in all of the..” to “maternal transmission of microbes to the F1s in all of the..” in the revised manuscript (Line number: 306-307).

Comment 43: Line 289: Once again, in several places in the manuscript there is a font change.

Response 43: As we mentioned in Response 15, we have made every effort to maintain the same font in the revised version of the manuscript.

Comment 44: Line 298: And also caesarean birth, correct?

Response 44: Thank you for pointing it out. We have revised the sentence by including “caesarean section birth” in the revised manuscript (Line number: 323).

Comment 45: Line 298-301: I find this sentence somewhat misleading as very few strains were observed in early life of the infant and maintained at infancy, at least according to Table S3 and Figure 1.

Response 45: Thank you for pointing it out. Since Figure 1 and Table S3 have been revised as requested by the reviewer, we have accordingly included this information in the Discussion section of the revised manuscript (Line numbers: 324-326).

Comment 46: Line 304-306: How does appearance of strains during pregnancy suggest changes in observed strains seen postpartum for the mother and during early life of the infant?

Response 46: Please note that there were no infant samples available for the pregnancy microbiome data set (Goltsman et al.). Also, no pregnancy samples were available for the mother-infant data set (Shao et al.). Therefore, we have included a caveat of sampling in the Discussion section of the revised manuscript (Line numbers: 331-332).

Since we were not able to investigate the direct consequences of having new strains in mothers during pregnancy, we made use of an animal model. Most importantly, this model allowed us to eliminate the impact of environmental microbes by using mice bred entirely in a gnotobiotic facility.

Therefore, we have also included this information in the Discussion section of the revised manuscript (Line numbers: 332-334).

Comment 47: Line 326-332: I think the authors are missing a critical component, the diet of mother and infant, that most likely plays a key role in structuring fecal bacterial communities. This should be discussed.

Response 47: We are unsure as to what the reviewer is referring to with this comment. In the revised manuscript, we have pointed out that the mice (mothers and infants) were all fed an autoclaved mouse chow diet to eliminate any environmental microbes.

The reviewer correctly states it is known that the gut microbial composition in the human infant changes as the infant transitions from breast milk to solid foods. However, the focus of our study was to examine the relationship between the microbe strains between the mother and infant (i.e. strain tracking for related/unrelated strains). In our opinion, a discussion on the “structuring of bacterial communities” would need to take into account individual differences in transition to solid food and individual diets that would be out of the scope of our study.

Comment 48: Line 338: Should “from caesarean birth” instead be “following caesarean birth”?

Response 48: As suggested by the reviewer, we have changed this sentence from “to infant from caesarean birth” to “to infant following caesarean birth” in the revised manuscript. (Line number: 365-366)

Comment 49: Line 348: This reviewer appreciates the authors acknowledging caveats of the study. In particular, it is noted that sampling of the F1 progeny’s feces was done at time of weaning. Please remind the readers as to when the dams were sampled as well. I think there would have been about a 7-day difference between the two sampling, correct?

Response 49: We clarified the times for sample collections in the Material and Methods section of the revised manuscript (Line numbers: 120-121). We have also discussed a limitation that F1 siblings might have shared microbes through coprophagy (Line numbers: 376-377).

Comment 50: Line 426: Segata reference is missing additional information
Segata N. 2018. On the road to strain-resolved comparative metagenomics. *mSystems* 3:e00190-17.

Response 50: Thank you for pointing it out. We have added “:e00190-17” information to the Segata reference (Line number: 460).

Comment 51: Line 440: “microbial” is misspelled.

Response 51: We have corrected the “microbial” word in the Reference section of the revised manuscript (Line number: 473).

Comment 52: Line 580: What was the cutoff for “abundant species”?

Response 52: As mentioned in Response 10, we have selected all species that had a cut-off value and were able to provide a WSS score for our FMT data set. We have accordingly revised Figure 5 along with legend (Line numbers: 604).

REVIEWER 2

Comment 1:

The analyses are elegantly done, and although the results are not groundbreaking, they elegantly confirm that engraftment in new niches brings microbes that are not observed in the source, because they were in very low load (rare).

My only observation is that the text of their manuscript has inconsistent fonts. I am sure this is fixable by the journal editors.

Response 1: Thank you. We have made every effort to maintain the same font in the revised version of the manuscript.